# Westdrive X LoopAR: An Open-Access Virtual Reality Project in Unity for Evaluating User Interaction Methods during Takeover Requests

**DOI:** 10.3390/s21051879

**Published:** 2021-03-08

**Authors:** Farbod N. Nezami, Maximilian A. Wächter, Nora Maleki, Philipp Spaniol, Lea M. Kühne, Anke Haas, Johannes M. Pingel, Linus Tiemann, Frederik Nienhaus, Lynn Keller, Sabine U. König, Peter König, Gordon Pipa

**Affiliations:** 1Institute of Cognitive Science, University of Osnabrück, 49090 Osnabrück, Germany; fnosratnezam@uos.de (F.N.N.); nmaleki@uos.de (N.M.); pspaniol@uos.de (P.S.); lkuehne@uos.de (L.M.K.); ankhaas@uos.de (A.H.); jpingel@uos.de (J.M.P.); litiemann@uos.de (L.T.); fnienhaus@uos.de (F.N.); lykeller@uos.de (L.K.); sabkoeni@uos.de (S.U.K.); pkoenig@uos.de (P.K.); gpipa@uos.de (G.P.); 2Center of Experimental Medicine, Department of Neurophysiology and Pathophysiology, University Medical Center Hamburg-Eppendorf, 20251 Hamburg, Germany

**Keywords:** VR research, out-of-the-loop unfamiliarity (OOTLU) autonomous driving, human–machine interaction, takeover request (ToR)

## Abstract

With the further development of highly automated vehicles, drivers will engage in non-related tasks while being driven. Still, drivers have to take over control when requested by the car. Here, the question arises, how potentially distracted drivers get back into the control-loop quickly and safely when the car requests a takeover. To investigate effective human–machine interactions, a mobile, versatile, and cost-efficient setup is needed. Here, we describe a virtual reality toolkit for the Unity 3D game engine containing all the necessary code and assets to enable fast adaptations to various human–machine interaction experiments, including closely monitoring the subject. The presented project contains all the needed functionalities for realistic traffic behavior, cars, pedestrians, and a large, open-source, scriptable, and modular VR environment. It covers roughly 25 km^2^, a package of 125 animated pedestrians, and numerous vehicles, including motorbikes, trucks, and cars. It also contains all the needed nature assets to make it both highly dynamic and realistic. The presented repository contains a C++ library made for LoopAR that enables force feedback for gaming steering wheels as a fully supported component. It also includes all necessary scripts for eye-tracking in the used devices. All the main functions are integrated into the graphical user interface of the Unity^®^ editor or are available as prefab variants to ease the use of the embedded functionalities. This project’s primary purpose is to serve as an open-access, cost-efficient toolkit that enables interested researchers to conduct realistic virtual reality research studies without costly and immobile simulators. To ensure the accessibility and usability of the mentioned toolkit, we performed a user experience report, also included in this paper.

## 1. Introduction

What defines the user-friendly design of automated systems has been the subject of scientific discussion for decades [1,2]. Especially in the upcoming years, when automated vehicles of SAE (society of automotive engineers) automation levels 3 and 4 will emerge, the demands on the driver’s cognitive system will alter radically, as the role of humans as continuously active decision-makers in vehicles is replaced by automated systems [3,4]. Such techniques include the Audi traffic jam pilot [5] or Tesla’s full self-driving beta [6]. Airlines’ experiences, where automated systems are already widely integrated, clearly state that such systems’ safety and reliability cannot be achieved by optimizing technical components alone [7]. Instead, the reliability of highly automated systems is primarily determined by the driver’s cognitive processes, meaning how fast a safe transition to manual drive is possible [8].

The need for a fast and safe transition applies particularly to situations where humans have the task of taking over system control in the event of sensor failures or malfunctions [9,10]. Thus, investigating the fluent integration of the takeover request (ToR) is crucial for the safety of any system with even partially automated driving features [11]. During a takeover request, the human driver most likely has to take over control in under 10 s, even when not engaged in driving-related activities [12,13,14]. Naturally, an orientation phase follows as the human driver has to assess the traffic situation [14]. Unfortunately, the driver’s reaction is often too slow in critical situations, potentially resulting in an accident in the small time frame (<4 s) before an impact occurs [15,16]. Even in the case of fast reactions within a time frame under 10 s, studies with prolonged driving have shown hectic responses by human drivers, which of course neither improved the reaction time nor the situational outcome [17,18].

This manuscript presents a new toolset for human–machine interaction research apart from typical screen-based simulators. Existing simulators are often based on actual car interior designs. Therefore, they offer only limited possibilities for human–machine interaction (HMI) research [19]. A very similar problem is posed by research on prototype cars in the real world, where realistic accident scenarios are costly and can only be generated to a minimal extent without endangering the test person involved. The project, called LoopAR, provides not only all the needed assets and an environment but also all the needed code to display the information of a takeover request as a freely programmable augmented reality (AR) feature in the windshield. The developed HMI displays the takeover request and highlights critical traffic objects to enable participants to take over more quickly and precisely. Our research is aimed toward safe and effective communication between car and driver. This is not only beneficial in terms of safety for the passengers but could also increase customer acceptance of highly automated vehicles, since up until now, malfunctions have been vital concerns of possible customers [20]. Since LoopAR is based on the project Westdrive [21], all the code needed and designed scenes are available in a Github repository. Project Westdrive is an open science VR project that tries to enable many researchers to conduct VR studies. It provides all the necessary code and assets in a public repository to set up VR studies. LoopAR is an extension of the Westdrive toolkit, focusing on the human–machine interaction. To fully use the project presented here, only a powerful computer, VR glasses, a simulation steering wheel and pedals, as well as Unity as a development program are required.

## 2. Methods and Main Features of LoopAR

The main focus of the presented project is versatility and modularity, which allows the fast adjustment of the environmental and functional objects via prefab and the provided code in the toolkit. Research on the interactions between humans and cars is mostly done with stationary simulators. Here, a whole car chassis is used, or only the interior is set inside a multi-screen setup. However, these classical setups are often expensive, and adjustments or graphical improvements of the stimuli used in an experiment are often not possible [22]. In the past few years, there has been a significant shift in research toward virtual environments. This is reflected by applications like Cityengine and FUZOR [23,24] and by the software for driving environments [25]. Still, experimental designs on human–machine interaction, in terms of specific car interior adjustments, are not possible yet. Therefore, the presented project enables the user to create experimental conditions and stimuli freely. All functionalities that are mentioned in the following are independent and can be adjusted at will. Additionally, the presented project does not need a specific hardware setup, making it easily adaptable and future-proof. New components, e.g., new GPUs and new VR devices, can be easily integrated into the setup displayed in Figure 1. The current requirements only apply to the VR devices used and are not bound to the toolkit. The following figure depicts an overview of the default experimental procedure, environmental structure, and data flow of the toolkit. Again, all of these defaults can be adjusted at will. The configurations presented here are intended to allow for a quick adaptation to other experiments.

### 2.1. Platform

Project LoopAR is made with the Unity editor 2019.3.0f3 (64bit). This software is a widely used game engine platform based on C# by Unity Technologies, supporting 2D, 3D, AR, and VR applications. The Unity editor and the Unity Hub run on Windows, Mac, and Linux (Ubuntu and CentOS), and built applications can be run on nearly all commercially usable platforms and devices. Unity also provides many available application programming interfaces and is compatible with numerous VR and AR devices [26].

The backend code of the project LoopAR was developed entirely using C# within Unity3D Monobehaviour scripting API. The backend comprises functionalities including dynamic loading of the environment, AI car controls, pedestrian controls, event controls, car windshields augmented reality controller, data serialization, and eye-tracking connection. Additionally, the presented project contains a C++ library enabling the force feedback for Microsoft DirectX devices that enables various force feedback steering wheels to function as controllers altogether. LoopAR code has been developed with modularity in mind to avoid complicated and convoluted code. All functionalities can be enabled or disabled individually using the Unity editor’s graphical interface based on need.

### 2.2. Virtual Environment

To test human–machine interactions, an interactive and realistic 3D environment is needed. LoopAR aims at a fully immersive experience of a highly automated car encountering critical traffic events. To be able to investigate different driving conditions and scenarios, we created four independent scenes. In the following section, the environment design decisions are presented together with a short description of the experimental scenes.

The LoopAR environment is based on real geographical information of the city of Baulmes in the Swiss Alps. We selected this region due to its variety of terrain, including a small village, a country road, a mountain pass, and a region suitable for adding a highway section, totaling around 25 km^2^ of environment and an 11 km continuous drive through different roads. To reduce the computational demands, we sliced the terrain into four areas. Due to the road network design, these separate environments can be merged (see Figure 2). These areas demand different driving skills from an automated driving vehicle and a human driver, reacting in different situations with different conditions according to the landscape and traffic rules. To make the region accessible in Unity, we used the collaborative project OpenStreetMap (OSM) [27] and the open-source 3D software Blender [28].

OpenStreetMap is a project with the aim of creating a free map of the world. It collects the data of all commonly used terrains on maps. The project itself collects information, so the data are free of charge. The virtual environment contains a mountain road scene (see Figure 3a), including curvy roads winding through a forest and steep serpentines running down a mountain. These curvy roads require various driving speeds (from 30 km/h or slower, up to 100 km/h on straight stretches). The overall traffic density is low.

The second area of the environment is the village “Westbrück” (See Figure 3b). Here, it is possible to test events in a more inhabited environment. This environment is characterized by narrow streets and dense traffic in low-speed environments.

The third scenario is the country road scene (see Figure 3c), designed for medium to high speed (~70 km/h), medium traffic density, and a long view distance. 

The last scenario for the participants is the highway scene (see Figure 3d), enabling critical traffic events with a higher speed and a low to medium traffic density. 

### 2.3. Critical Traffic Events

To test the participant’s behavior in critical traffic events, we created limited event zones, where the monitoring of a participant can be achieved in a well-controlled environment. In Figure 4, one example of a traffic event is displayed. Each environment (mountain road, city, country road, and autobahn) has three critical traffic events. These zones are the core of the possible measurements in the presented toolbox. Simply put, the event system is realized by a combination of several trigger components. These independent triggers are activated when the participant enters the start trigger (Figure 4: green gate). The event zone is restricted within “boundary” triggers (Figure 4: yellow boxes). These triggers get activated on contact, which is considered a participant’s failure. Contact with the event triggers leads to a black screen followed by a respawn of the car at a point after the event (Figure 4: pink box) and giving back the car’s control. An event is labeled as “solved” when the participant enters the end trigger (Figure 4: red gate) without crashing, i.e., making contact with the “boundary” triggers. All critical events can be adjusted at will, and a prefabricated file is stored in the repo to create new events. The triggers are all visible in editor mode but invisible to the participant.

### 2.4. Cars and Traffic Behavior

Within the event zones, dynamic objects, such as other road users, are needed to create realistic traffic scenarios. The repository presented here contains various animated pedestrians, animals, and cars to create a broad range of critical situations. Additionally, there are some busses and trucks, and some construction site vehicles that can be used. Furthermore, a user’s own fbx models, as well as vehicles from the Unity asset store, can be added. For more details, please see the Appendix A. All cars used are based on the Unity wheel collider systems of the Unity3D physics engine. In the Car Core Module, user input is translated into the motor control of the participant’s car. The input consists of the motor torque, brake torque, and steering, which are applied to the wheels. This functionality is called AI control. It allows a seamless transition from automated to manual driving when activated. To facilitate realistic traffic behavior, an additional AI module enables cars to follow predefined paths. Paths followed by AI Cars and walking pedestrians were defined by mathematical Bézier curve paths [29], which were realized by the Path-creator tool [30]. Speed limit triggers inside the scene manipulate the AI’s aimed speed, handling the input propagated to the Car Core Module. Another module of the car AI allows the AI cars to keep a distance from each other. The goal is to create an easily configurable and interchangeable traffic AI for multiple study designs. With these measures, we maximized the car physics and traffic simulation realism while ensuring easy adjustments.

### 2.5. Experiment Management

Data sampling, dynamic objects, and driving functionalities within the event zones are controlled by a system of experiment managers that handle scene-relevant information and settings shortly before and during the real experiment phase. It handles different camera settings, the information given by triggers inside the scene, and the participants’ respawn in case of failure. Before an experiment starts, initial adjustments start the experiment. These adjustments configure the experiment to the participant and include the eye calibration, eye validation, seat calibration, and a test scene.

The eye-tracking component in this setup comprises an eye-tracking calibration, validation, and online gaze ray-casting, which can record necessary gaze data during the experiment. The component was built for the Tobii HTC Vive Pro Eye device but is intended to keep the VR component interchangeable. It was designed as a simple connector to tap into SRanipal and the Tobii XR SDK (see Figure 5). The eye calibration is performed with the built-in Tobii eye calibration tool. The validation is set in the corresponding validation scene, which provides a simple scenario with a fixation cross. Validation fails if the validation error angles exceed an error angle of 1.5° or the head is moved by 2” from the fixation cross. During the experiment, the eye orientation, position, and collider hits are stored with a calculated gaze ray of both eyes. Currently, it is set to receive information about any object inside these rays to prevent the loss of viable information by objects covering each other.

In addition to the eye-tracking data, input data of the participant as well as scene-relevant information, such as the number of failed critical traffic events, are saved using generic data structures and Microsoft Linq, serialized into JavaScript object notation (JSON), and saved with a unique ID at the end of each scene. The generic data structure used in the project ensures flexibility, as different data types can be added or removed from the serialization component. This approach guarantees the highest compatibility with varying analysis platforms such as R or Python for the data gathered with LoopAR.

By conducting data saving, and given the nature of the experimental setup, we aim for a stable and high frame rate to provide a less sickness-inducing experience. A stable visual experience can be seen as a prerequisite to avoid potential sickness [30,31]. The desired optimum for the experiments is a stable frame rate matching the fixed rate of 90 Hz used by the manufacturers HTC and Oculus. Our current frame rate in the different scenes yields an average of 88 samples per second in our test setup, matching the maximum sampling rate of the HTC Vive with 90 fps.

## 3. Hardware Requirements

The setup used and presented here is thought to be a cost-efficient and very mobile replacement for maintenance-intensive, rigid, and expensive driving simulators for studies on human behavior in the context of self-driving cars. A key advantage is freedom regarding the selected components. The only requirement for operation is granting the computing power for the entire system, which consists of a core setup only of a computer, a head-mounted display, and a steering wheel (see Table 1).

As a virtual reality device, we used the HTC Vive Pro Eye with an integrated Tobii Eye Tracker. It is a cable-bound head-mounted display that enables the participant to transfer movements into virtual reality. Although we are using the Vive Pro exclusively at our department, the LoopAR experiment is not dependent on this specific VR device. We used the components of the setup with 90 fps sampling and display.

## 4. Discussion

In the presented paper, we describe LoopAR as a modular toolkit to test a takeover of control in critical traffic situations from automated cars to human drivers by combining VR and eye-tracking in an interactive and immersive scenario. Its current state and design provide a promising, new, low-cost, and mobile setup to conduct studies that were traditionally only done in stationary simulators. The current code, as well as the 3D environments, can be adjusted at will. With newly implemented code, it is not only possible to simulate a large and highly realistic VR environment, but it is also possible to create a broad range of applications in VR research that is not only bound to HMI investigations. A large part of the assets used are from Unity’s asset store and the 3D platforms Sketchfab and Turbosquid. Therefore, it is possible to change the number, size, and shape of all objects in each scene.

All of the functionalities above, and assets presented here, are under constant improvement. By writing, five new projects, ranging from ethical decision-making over EEG implementation to human spatial navigation, arise from the presented toolkit, which will also develop new assets and features implemented into the toolkit later on. The authors want to emphasize the modularity and adaptability of this VR toolkit.

## 5. User Reports

To check for the user friendliness of the presented toolkit, a System usability score (SUS)-based report was performed [32]. Here, we asked 11 of the current users between the age of 23 and 34 (5 female) to evaluate the usage of the main features in the toolbox starting from cloning the repository, adjusting the environment, and manipulating dynamic objects in an example scene. While doing so, we asked the participants to evaluate the feasibility of the tasks. User experience in Unity and C# programming varied from no experience to expert levels with more than 3 years of experience. Our top findings, depicted in indicate that the toolbox is perceived as well documented, and advanced Unity users faced no major problems building and altering their project created with this toolbox (see Figure 6). While some steps in the procedures might be challenging to new users, the Westdrive X LoopAR toolbox seems to be a useful foundation for all users.

As about 90% of the users stated that there was no need for previous knowledge to use this toolkit, we feel supported in our claim for sufficient usability. The fact that no user had any difficulty in cloning the repository and/or using the basic scene management furthermore fortifies our efforts. Most problems occurred in the tasks about data saving and the manipulation of objects related to critical traffic events. These tasks required more technical and programming knowledge. Therefore, we will focus on simplifying these tasks in future versions.

We would like to conclude that this toolkit proved to be useful for the creation of VR experiments. All users over all levels of experience were able to access and open the toolkit. Moreover, all users were able to navigate through the project structure and found the important assets. To further improve the toolkit, we will create more tutorial videos, since these received very positive feedback from all the participants. Moreover, we will implement the more complex features related to experimental procedures into the Unity editor to ease its use, especially for less experienced users.

## 6. Conclusions

This article describes a new virtual reality toolkit for Unity applications investigating human–machine interaction in highly automated driving developed by us. The presented setup is thought to be a mobile, cost-efficient, and highly adaptable alternative to chassis simulators that closely monitor the participants. It is particularly noteworthy that there is not only a drastic reduction in costs but also an improvement to the adaptability of the software as well as the used hardware. All components are fully upgradable, in case there are better products in terms of image quality or computing power. LoopAR allows interested researchers to conduct various virtual reality experiments without creating the needed environment or functionalities themselves. For this, we have provided an area of almost 25 km^2^ based on OSM data. The toolkit presented here also includes all the necessary assets and basic prefabs to quickly and precisely create a wide variety of virtual environments. Additionally, the LoopAR toolkit contains components of the experimental procedure and data storage.

## Figures and Tables

**Figure 1 sensors-21-01879-f001:**
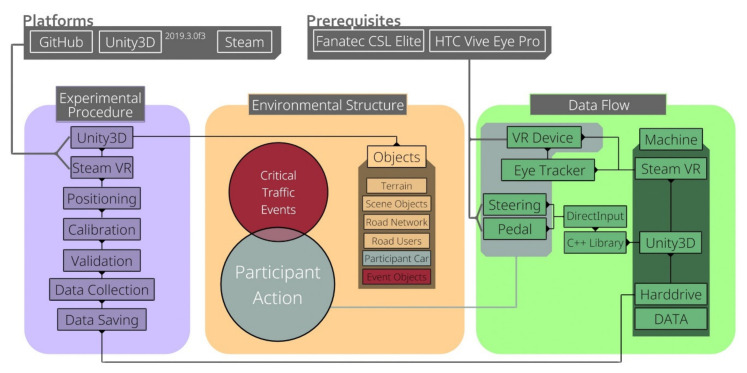
A simplified overview of the toolkit structure. It includes the default experimental procedure, a possible example of how the environmental structure can be used, and the standard data flow of the toolkit.

**Figure 2 sensors-21-01879-f002:**
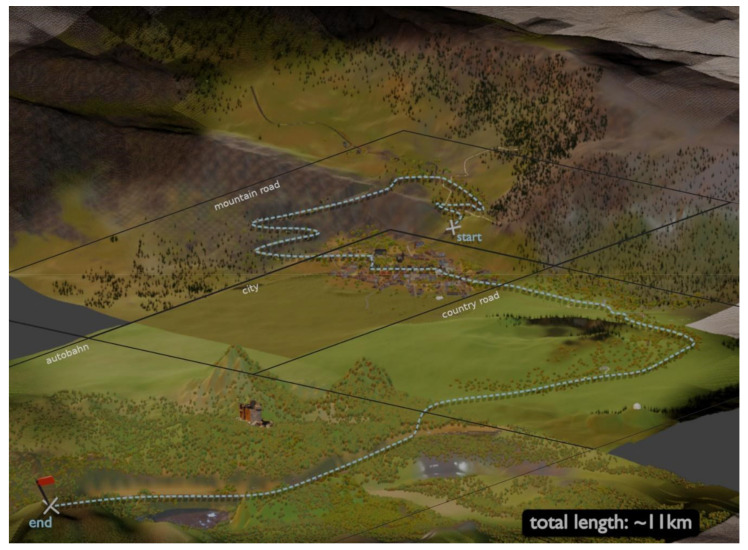
LoopAR map preview: mountain road (3.4 km), city (1.2 km), country road (2.4 km), and highway (3.6 km).

**Figure 3 sensors-21-01879-f003:**
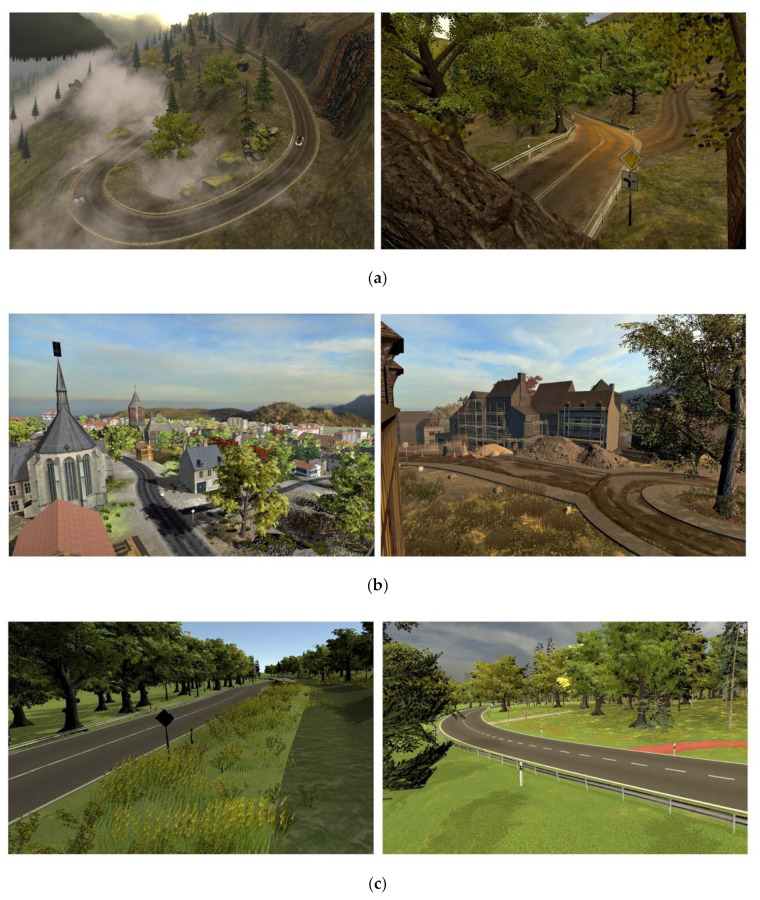
(**a**) Pictures of the different scenes from the mountain road. (**b**) Pictures of the different scenes from the village “Westbrück”. (**c**) Pictures of the different scenes from the country road. (**d**) Pictures of the different scenes from the highway.

**Figure 4 sensors-21-01879-f004:**
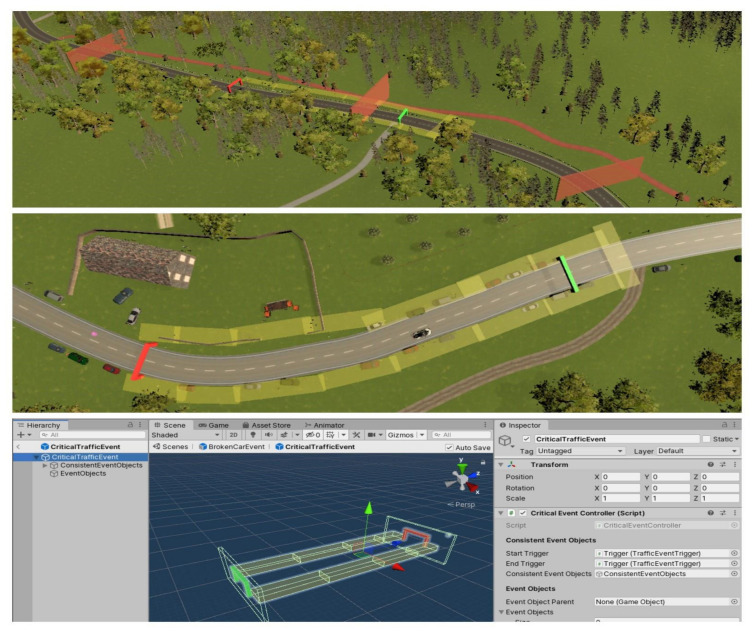
Critical traffic event prefab and its implementation.

**Figure 5 sensors-21-01879-f005:**
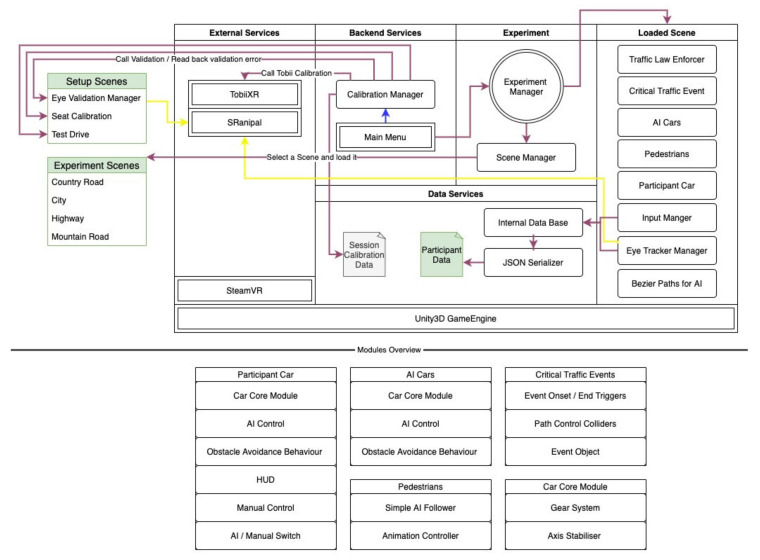
Scheme of the LoopAR functionalities and components illustrating the interaction of the different services and manager scripts within the Unity environment.

**Figure 6 sensors-21-01879-f006:**
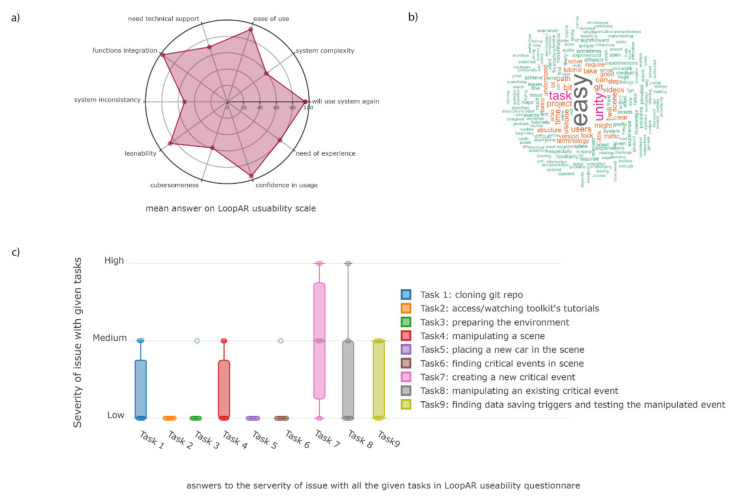
Visualization of the usability report items: (**a**) a radar plot of the system usability scale data; (**b**) a word cloud showing most frequently used words in the comments; and (**c**) a severity of issue bar plot, related to the tasks in the usability report. Low equals no delay in time or perceived obstacles, medium refers to a completed task with added effort. High indicates noticeable delay or frustration and that the participant may not be able to complete the task.

**Table 1 sensors-21-01879-t001:** An overview of the used components in the LoopAR setup. The steering wheel and the VR device are only suggestions in this listing. The LoopAR toolkit also works with other devices.

GPU	Nvidia GeForce RTX 2080, equivalent or better
CPU	Intel(R) Xeon E5-1607 v4, equivalent or better
RAM	32 GB
Video Output	HDMI 1.4, DisplayPort 1.2 or newer. USB port, 1x USB 2.0 or better
Operating System	Windows 10
VR HMD	Vive Pro Eye with built-in Tobii Eye Tracker
Steering Wheel	Game-ready Fanatec CSL Elite Steering Wheel and pedals

## Data Availability

The datasets for the asset foundation and scripts can be found in the Westdrive repository https://github.com/Westdrive-Workgroup/MotorCity-Core (accessed on 22 February 2021). The dataset for the LoopAR project is accessible at: https://github.com/Westdrive-Workgroup/LoopAR-public (accessed on 22 February 2021).

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
