# Peer review of "Westdrive X LoopAR: An Open-Access Virtual Reality Project in Unity for Evaluating User Interaction Methods during Takeover Requests"

_sensors, 2021, doi:10.3390/s21051879_

Round 1

Reviewer 1 Report

This paper presents a modular toolkit to test critical traffic situations in autonomous driving. It could be improved if the authors provide at least a user experience report. It would be much scientific solid if the authors show how to utilize their modular to VR environment, such as the experimental results that apply to VR researches. Although the authors insist that their work is effective to adapt users to the VR environment, there is no clue to demonstrate it.

Author Response

Number

Reviewer

Comments

Answer

1

1

It could be improved if the authors provide at least a user experience report

Dear Reviewer 1, first of all,  thank you very much for your comments and we addressed the issue. We are not sure how we could include a UX report into this technical note. A whole report might exceed the scope of the manuscript quickly. As a solution, we gathered experiences from current users and wrote a section in the discussion (2nd paragraph in the discussion)

2

1

It would be much scientific solid if the authors show how to utilize their modular to VR environment, such as the experimental results that apply to VR researches.

Thank you very much for this hint. We agree that this would be beneficial to show how to utilize the environment. Therefore, we will add a tutorial video into the git repository as well as into the supplementary files where we will describe how to clone the repo and how to use its main features.

3

1

Although the authors insist that their work is effective to adapt users to the VR environment, there is no clue to demonstrate it.

Also thanks for that. We will try to incorporate the adaptation into the videos. Also this is reflected in the users experience paragraph.  The video tutorial will include cloning and how to utilize the main features. 

Reviewer 2 Report

The article is good and well written. I suggested that the authors include in section 2 (methods) a simple flowchart to improve the reader's comprehension. Also, Figure 4 is missing in the article. Place components among lines 221 and 222 as a table e named the table.

Some questions: the routes can be located in any place in the world? What about heavy vehicles? Are they can be included in the analysis?

Please, enlargest the Figure 3. Provide the best link to show supplementary materials.

Author Response

4

2

I suggested that the authors include in section 2 (methods) a simple flowchart to improve the reader's comprehension.

Dear reviewer 2, thank you very much for your valuable comments. We included a flowchart in the method section

5

2

Also, Figure 4 is missing in the article.

Done. We included the missing figure in line 185. Thanks alot for the hint.

6

2

Place components among lines 221 and 222 as a table not only as inlines. Also, name this table.

Addressed. Thank you very much for the input. We provided a table with a description in lines 221 and 222

7

2

Some questions: the routes can be located in any place in the world?

Yes - that’s a very good question. OpenStreetMap is a project with the aim of creating a free map of the world. It collects data about roads, railways, rivers, forests, houses and everything else that is commonly seen on maps. Data is collected by the project itself, so it is open access. Anyone may use the OpenStreetMap data free of charge and process it as they wish. We put a short sentence into the manuscript to clarify the use of OSM.

8

2

What about heavy vehicles? Are they can be included in the analysis?

Also this is addressed in the Manuscript. Thanks a lot for asking this. And yes! We do also have heavy vehicles in our toolbox. There are some busses and trucks and some construction site vehicles that can be used. They are not mentioned, because we do not have a particular focus on heavy vehicles, but they can be used. Additionally, all fbx models of own heavy vehicles and also vehicles from the Unity asset store can be added.

Reviewer 3 Report

Summary

A virtual reality toolkit is described which the authors present as being particularly suitable for investigating the type of human-machine interaction that may occur in highly automated driving. The toolkit is depicted as a low-cost alternative to professional driving simulators. 

Evaluation

The authors describe an interesting toolkit which may be of use to a growing number of researchers in the area of automated driving. With a few changes (see below) the paper should be publishable.

Comments

(1) The paper should be limited to describing the tool and the authors should strictly avoid any sort of self-praise and advertising language. For instance, it is in appropriate to advertise the toolkit as one which “is thought to fundamentally change research” (line 68). Such evaluations must be left to others and this one is particularly over the top and inappropriate. Similarly, advertising language such as “this project quickly enables users” (line 228) and “it can have a significant influence” (line 229) must be deleted. 

(2) Line 4, “TOR”: Abbreviations must be avoided as much as possible because they reduce readability while saving only negligible amounts of space. Abbreviations in headers are particularly detrimental and this one is even worse in that is it inconsistent in spelling with the main text (e.g., line 50: “ToR”). Please replace abbreviations by real words which the average reader can easily understand.

(3) Line 20: Replace “We developed” by “Here we describe”.

(4) Line 71: Readers will typically have no idea what “WestDrive” is (and may have trouble locating it given the many different spellings of the term on Osnabrueck University’s web pages [Westdrive, West Drive,…] and given the fact that the term is far from unique). Please describe what “WestDrive” is in a short paragraph in the paper. 

(5) Line 147: This section is less accessible than other sections and should be illustrated with an example.

(6) Line 155: A “scenario” cannot be “successful”.

(7) Line 185: Figure 4 is missing.

(8) Line 206: It is inappropriate and misleading to write that “stable and high frame rate … avoids motion sickness.” Motion sickness is primarily caused by a mismatch between visual and vestibular information (or predicted sensory input) in the brain and a large number of moderator variables also play a role (air quality, gender, training…). In this sense VR environments such as the one described here are typically much worse (in that they cause motion sickness much more readily) than professional driving simulators. The inappropriate phrase must be deleted and replaced by a tenable statement about motion sickness and the disadvantage of VR environments compared to professional driving simulators.

(9) Line 221f: “Intel(R) Xenon RE5-1607” should probably be replaced by “Intel(R) Xeon E5-1607”. There should also be no dash preceding the “Eye”.

(10) Line 223: “In the presented paper, we developed” should be replaced by “In the present paper, we described”.

(11) Line 239: Sentences such as “We are happy to present LoopAR, and we are already looking forward to the many extensions of the WestDrive project” may be in order in an oral presentation but they are inappropriate in a journal publication. 

(11) The authors should refer readers to a scientific publication in which the toolkit described here was used successfully. It is important for readers to know whether such a proof of the utility of the toolkit exists. 

Author Response

9

3

(1) The paper should be limited to describing the tool and the authors should strictly avoid any sort of self-praise and advertising language. For instance, it isn’t appropriate to advertise the toolkit as one which “is thought to fundamentally change research” (line 68). Such evaluations must be left to others and this one is particularly over the top and inappropriate. Similarly, advertising language such as “this project quickly enables users” (line 228) and “it can have a significant influence” (line 229) must be deleted. 

Dear Reviewer 3, first of all, thank you very much for the detailed comments on our manuscripts. We find your comments very helpful and think they definitely improve the manuscript.  We tried to incorporate all of your suggestions to your liking.

Done. We took out the inappropriate formulations in line 68, as well as in line 228 and 229.

10

3

(2) Line 4, “TOR”: Abbreviations must be avoided as much as possible because they reduce readability while saving only negligible amounts of space. Abbreviations in headers are particularly detrimental and this one is even worse in that is it inconsistent in spelling with the main text (e.g., line 50: “ToR”). Please replace abbreviations by real words which the average reader can easily understand.

Agreed. We fully agree with your concerns regarding the abbreviations. We changed all abbreviations in the text. 

11

3

(3) Line 20: Replace “We developed” by “Here we describe”.

Thank you for the hint, we changed the abstract text as you suggested. 

12

3

(4) Line 71: Readers will typically have no idea what “WestDrive” is (and may have trouble locating it given the many different spellings of the term on Osnabrueck University’s web pages [Westdrive, West Drive,…] and given the fact that the term is far from unique). Please describe what “WestDrive” is in a short paragraph in the paper.

Done. Thanks for the input. We made the spelling consistent in the manuscripts.

We also added a small paragraph of what project westdrive

Also, we are open for suggestions regarding project names. If you have any suggestions for a more precise and unique name, please let us know.

13

3

(5) Line 147: This section is less accessible than other sections and should be illustrated with an example.

Addressed. We understand that this might be a bit abstract since we are talking about objects on our route. We tried to refer to the according figures earlier in the text to illustrate this.

14

3

(6) Line 155: A “scenario” cannot be “successful”.

This is correct. We changed the formulation to: An event is labeled as “solved”.

15

3

(7) Line 185: Figure 4 is missing.

Done. We inserted the missing figure as an overview of all components.

16

3

(8) Line 206: It is inappropriate and misleading to write that “stable and high frame rate … avoids motion sickness.” Motion sickness is primarily caused by a mismatch between visual and vestibular information (or predicted sensory input) in the brain and a large number of moderator variables also play a role (air quality, gender, training…). In this sense VR environments such as the one described here are typically much worse (in that they cause motion sickness much more readily) than professional driving simulators. The inappropriate phrase must be deleted and replaced by a tenable statement about motion sickness and the disadvantage of VR environments compared to professional driving simulators. 

This is true. We changed the text in the manuscript accordingly

17

3

(9) Line 221f: “Intel(R) Xenon RE5-1607” should probably be replaced by “Intel(R) Xeon E5-1607”. There should also be no dash preceding the “Eye”.

Done. Thanks also for this, we changed it in the manuscript

18

3

(10) Line 223: “In the presented paper, we developed” should be replaced by “In the present paper, we described”.

Done as well.

19

3

(11) Line 239: Sentences such as “We are happy to present LoopAR, and we are already looking forward to the many extensions of the WestDrive project” may be in order in an oral presentation but they are inappropriate in a journal publication. 

Agreed.  We deleted all of these sentences.

20

3

(12) The authors should refer readers to a scientific publication in which the toolkit described here was used successfully. It is important for readers to know whether such a proof of the utility of the toolkit exists.

There are no publications up to now using the toolkit. However, there are new experiments ready for data acquisition, that are based on the here presented toolkit. They range from ethical evaluation of driving actions, to social interaction in VR spatial representation and use nearly all components and modules of LoopAR. Sadly, due to the current lockdown this is delayed.

Nevertheless,  we will include tutorial videos in the Github repository to recruit a larger user base.

Round 2

Reviewer 1 Report

This paper developed the open-access engines for VR studies. This paper would be more scientific solid if the authors could prove that the developed toolkit can be successfully applied to VR applications. 

Author Response

1

1

This paper developed the open-access engines for VR studies. This paper would be more scientific solid if the authors could prove that the developed toolkit can be successfully applied to VR applications.

Dear Reviewer 1,

We agree that a proof of the scientific usability of this toolkit would be beneficial for the paper. Therefore, we will include example videos of two different experiments in the repo.

Here, we will demonstrate complete trials of two current projects: The first one is an experiment on transparency and trust in highly automated cars, the other project is about ethical evaluations of driving actions (EEDA).

Both projects are using the presented toolkit as foundation and use most of its features.

The EEDA project is currently installed at the Deutsche Museum Bonn (https://www.deutsches-museum.de/bonn/information/aktuell/veranstaltungen-2021/mission-ki/). It’s ready for data acquisition and will start as soon as the circumstances allow so.

We hope that this proves that the toolkit can be successfully applied to VR applications.

Reviewer 3 Report

From my point of view, the paper is ready for publication. One hint: On p. 8 of 11 “Intel(R) Xenon” should be “Intel(R) Xeon”, not the chemical element. 

Author Response

1

3

From my point of view, the paper is ready for publication. One hint: On p. 8 of 11 “Intel(R) Xenon” should be “Intel(R) Xeon”, not the chemical element.

Dear Reviewer 3, thank you very much. The mistake is addressed.